# The Multi-Analytical Characterization of Calcium Oxalate Phytolith Crystals from Grapevine after Treatment with Calcination

Gwenaëlle Trouvé [1,2,*], Laure Michelin [2,3], Damaris Kehrli [1,2], Ludovic Josien [2,3], Séverinne Rigolet [2,3], Bénédicte Lebeau [2,3] and Reto Gieré [4]

1   Université de Haute-Alsace, GRE UR2334, F-68100 Mulhouse, France
2   Université de Strasbourg, F-67000 Strasbourg, France
3   Université de Haute-Alsace, CNRS, IS2M UMR 7361, F-68100 Mulhouse, France
4   Department of Earth and Environmental Science, University of Pennsylvania, Philadelphia, PA 19104-6313, USA
*   Correspondence: gwenaelle.trouve@uha.fr

**Abstract:** Calcium oxalate phytoliths are one of the most prominent types of Ca speciation in the plant kingdom, and they store extensive amounts of carbon in crystalline form. Ca phytoliths were investigated in the root, trunk, and bark of *Vitis vinifera Chasselas* from a vineyard in Alsace, France. A multi-analytical approach was used, which included SEM coupled with EDX spectroscopy, XRD, XRF, TGA, and $^{13}$C-NMR spectroscopy. These techniques revealed that phytoliths are composed of crystalline calcium oxalate monohydrate (whewellite). The whewellite crystals exhibited mostly equant or short-prismatic habits in all of the three studied grapevine parts, but bipyramidal crystals also occurred. Raphide crystals were only observed in the root, where they were abundant. Instead of using wet chemical procedures to extract the mineral components from the organic parts of the biomass, a thermal treatment via calcination was chosen. The suitable temperature of calcination was determined through TGA experiments. The calcination of the biomass samples at 250 °C enhanced the amounts of Ca phytoliths in the residual chars. The thermal treatment, however, affected the appearance of the Ca oxalate crystals by producing surfaces that displayed macroporosity and by creating fractures. For calcination at both 300 °C and 350 °C, Ca oxalate lost a molecule of carbon monoxide to form Ca carbonate, and the modifications of the original crystal surfaces were more pronounced than those observed after thermal treatment at 250 °C.

**Keywords:** phytoliths; calcium oxalate; calcination; SEM/EDX; XRD; $^{13}$C-NMR





## 1. Introduction

Calcium (Ca) is an essential nutrient for plants, where it plays an important role in a range of processes, including strengthening cell walls and membranes, balancing anions in vacuoles, protecting against stress, and intracellular signaling [1,2]. A large proportion of the total Ca concentration present in many plants occurs in the form of Ca oxalate crystals [3,4] which have not only been reported in all major groups of plants but also in other life forms, including humans [3,4]. In plants, Ca oxalate crystals are formed through a complex, highly regulated process, which occurs in specialized cells, crystal idioblasts [3]. Although the production of these crystals is genetically regulated, it is also influenced by environmental, physical, chemical, and biological conditions [5–7].

Ca oxalate occurs in various chemical forms: The monohydrate $Ca(C_2O_4) \bullet H_2O$, named whewellite, is the predominant mineral in the plant kingdom [8], and therefore, it represents one of the most abundant minerals at or near the Earth's surface [8]. Whewellite is also the principal component of human kidney stones [8]. Calcium oxalate further occurs as a dihydrate variety, $Ca(C_2O_4) \bullet 2H_2O$, known as weddellite, and in a trihydrate form,

$Ca(C_2O_4) \bullet 3H_2O$, named caoxite [5]. Because Ca oxalates are produced by living systems in such vast quantities, these minerals are able to sequester extensive amounts of atmospheric $CO_2$ in solid form [8]. This capability of storing $CO_2$ is analogous to that exhibited by carbonate minerals, such as calcite ($CaCO_3$) or dolomite ($CaMg(CO_3)_2$), which—unlike Ca oxalates—form entire geological units within the Earth's crust.

In plants, Ca oxalate crystals occur as phytoliths, i.e., solid mineral bodies. Their morphology, structure, and chemical composition, however, are distinct from another important class of phytoliths, namely opaline phytoliths. The latter, also occurring in many plant species, are present as rounded (bilobate or trilobate) or irregularly shaped, non-crystalline mineral bodies composed of opal, $SiO_2 \bullet nH_2O$ [9,10], and they play an important role in soil science, archeology, paleoecology, and climate science [11–13]. Ca oxalate phytoliths have several widely disparate functions, ranging from the stabilization of the plant structure and defense against herbivores to the regulation of calcium homeostasis, serving as an internal source of $CO_2$, and the detoxification of aluminum and certain heavy metals [3,14–16]. The abundance of Ca oxalate crystals has also been shown to fluctuate seasonally in some plants (e.g., grapevine), because the crystals act as a Ca reservoir and can supply this nutrient when required for plant growth, thus participating in plant metabolism [17].

Ca oxalate phytoliths are found in various parts of plants, for example, roots, stems, leaves, barks, seeds, and fruits, and they occur as intra- or extracellular deposits [3,5,7]. Morphologically, whewellite in plants exhibits a widely variable appearance, ranging from prismatic, tabular, and acicular habits of individual crystals to spherical aggregates (referred to as "druses") and clusters of tabular crystals, known as "crystal sand" [8,18,19]. Acicular crystals are characterized by sharply pointed ends and, in some cases, exhibit grooves along their sides; they may occur either as raphides, i.e., bundles of needles, or individually as styloids [18,19]. In addition, whewellite crystals vary tremendously in size and are frequently twinned, also in geological occurrences [7,18]. The morphological features of the Ca oxalate crystals are characteristic of different plant species and represent a taxonomic trait [7,18], and the presence or absence of crystals may be used for elucidating evolutionary relationships between plant species [6].

Silica phytoliths from several types of biomass may contain carbon inclusions whose origins are widely discussed and disputed in the literature [20,21], with some scientists emphasizing that this carbon represents a product of the sequestration of atmospheric $CO_2$ [22]. Within the current global warming context, this point of view is very important, as it could promote the selection of industrial crops that not only sequester extensive amounts of atmospheric $CO_2$ in the organic parts of the biomass but also in the form of minerals, i.e., phytoliths [23]. In China, for example, cultivation of certain rice species is favored due to the presence of carbon inclusions in Si phytoliths [23]. Oxalogenic plants, on the other hand, produce phytoliths that contain carbon as a major component (16.4wt% in the case of whewellite). Moreover, this Ca oxalate, when released to the soil upon the degradation of the host plant, represents a source of carbon, energy, and electrons for certain soil organisms (e.g., oxalotrophic bacteria), which metabolically transform it into Ca carbonate, thus increasing the long-term carbon storage capacity of soils [22]. Because Ca oxalates are so common in many plants, and because of the importance of the oxalate–carbonate pathway in soil carbon storage, it is important to acquire deeper knowledge regarding the abundance and characteristics of these phytoliths in various industrial crops.

The objective of this study was to test calcination as a method of dry extraction of Ca oxalate phytoliths from various parts of a grapevine for subsequent physicochemical characterization. Si phytoliths consist of silica, which is relatively insoluble at low pH and thus can be extracted and quantified through acid digestion. For the extraction of Si phytoliths from plant materials, various wet-chemical protocols exist, which digest the biomass or separate it from phytoliths by using liquids, such as hot acids, sodium hypochlorite solution, or alcohol [5,19,24–26]. Another approach is extraction using a dry method, called calcination. In our study, wet extraction was first tested, but the acids (HCl and $H_2O_2$) dissolved the Ca oxalate, and most of the Ca was transferred to the solution.

This investigation aimed at testing the effect of temperature on Ca oxalate phytoliths by characterizing oxalate crystals both before and after their extraction from the organic parts of the biomass. In addition, a multi-analytical approach was chosen to test the applicability of XRD and NMR spectroscopy as bulk techniques to both the original biomass and its calcination products. In a recent study, a similar approach has been applied to Ca oxalate in another type of biomass [27].

## 2. Materials and Methods

### 2.1. Starting Materials

Because live grapevines have a high economic value in the wine-growing region of Alsace, France, this study focused on a dead foot of a grapevine plant (Figure S1) collected during the winter period (January 2022). The grapevine corresponds to an American variety of Chasselas (*Vitis vinifera Chasselas*), harvested in the Region of Colmar (department of Haut-Rhin, France) by the LVBE Laboratory (Laboratoire Vigne, Biotechnologies et Environnement) of the University of Haute-Alsace. The grapevine sample was washed with large volumes of water for 15 min to remove soil particles and subsequently sun-dried outside for two weeks. The total mass of the grapevine was 4.1 kg. This sample allowed us to study three types of materials: the root, the trunk, and the bark. Root samples were taken from the part that was the deepest underground (Figure S1). The studied bark was the material separated from the trunk (Figure S1). All three types of materials were cut into small cubes. Due to the presence of soil minerals, the cubes were first washed in deionized water for 24 h, followed by ultrasonication for 1 h. Subsequently, the cleaned cubes were dried for 48 h in an oven at 105 °C prior to being ground to a grain size $\leq$1 mm in a knife grinder.

Calcium oxalate purchased from Sigma-Aldrich (CAS number 24804-31-7, specified as $C_2CaO_4 \bullet xH_2O$) was used as reference material. The characterization of this material using both thermogravimetric analysis (TGA) and X-ray diffraction (XRD) revealed that it is a monohydrated Ca oxalate form, but that it also contains traces of Ca hydroxide ($Ca(OH)_2$) (see below).

Three carbon-containing substances (cellulose, hydrolytic lignin, and xylan) were used as reference materials for $^{13}C$-nuclear magnetic resonance (NMR) spectroscopy. Cellulose and hydrolytic lignin were purchased from Aldrich, with CAS numbers 9004-34-6 and 8072-93-3, respectively. Xylan from birch wood was purchased from Fluka, with CAS number 9014-63-5, and it contains more than 95% of xylose.

### 2.2. Calcium Oxalate Extraction Protocols

With Ca oxalate being soluble in acids, wet digestion was not deemed the appropriate protocol. Therefore, in order to extract the Ca oxalate phytoliths for subsequent characterization, the calcination protocol was favored.

Calcination is a thermal treatment performed at high temperatures in order to decompose organic components, such as cellulose, xylan, and lignin polymers, present in biomass. The resulting residues can then be analyzed and characterized. The main minerals contained in biomass are accumulated in these residues, which are ash and/or chars depending on the calcination temperature [28,29].

To determine the temperature of calcination that should allow for the decomposition of biomass polymers without decomposing Ca oxalate phytoliths, TGA was performed on the reference Ca oxalate as well as on samples of the grapevine root, trunk, and bark, separately (Figure 1). The TGA experiments were performed under an airflow rate of 100 mL·min$^{-1}$ and with a temperature ramp of 5 °C·min$^{-1}$. The thermobalance was a Q500 from TA Instruments. The mass of each investigated biomass sample was 10–15 mg.

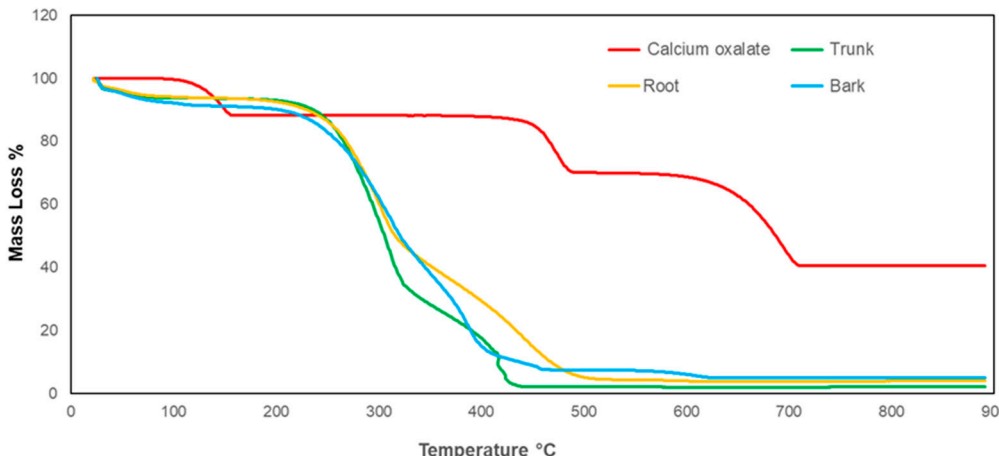

**Figure 1.** TGA curves of reference Ca oxalate and root, trunk, and bark of *Vitis vinifera*.

As shown in Figure 1, the first step of 12.3wt% between 100 °C and 200 °C (red curve) corresponds to the loss of one molecule of water, indicating that the purchased Ca oxalate from Sigma-Aldrich (labeled as $C_2CaO_4 \bullet xH_2O$) is a monohydrate form, where x = 1 (Reaction (1)). At higher temperatures, between 400 °C and 500 °C, a loss close to 22wt% points to the formation of Ca carbonate via the release of one molecule of carbon monoxide (Reaction (2)). The last step, the temperature range of 600 °C to 700 °C, represents the decarbonatization of Ca carbonate into lime (Reaction (3)). The corresponding reactions are as follows:

$$Ca(C_2O_4) \bullet H_2O \rightarrow Ca(C_2O_4) + H_2O \tag{1}$$

$$Ca(C_2O_4) \rightarrow CaCO_3 + CO \tag{2}$$

$$CaCO_3 \rightarrow CaO + CO_2 \tag{3}$$

These results are consistent with those previously reported by Ozawa [30]. Since the TGA analyses revealed that Ca oxalate decomposes above 400 °C (Figure 1), a thermal treatment via calcination is possible between 200 °C and 400 °C. The TGA curves of the studied biomass materials show two major weight losses between 200 and 450 °C, which correspond to the decomposition of organic components (hemicellulose, cellulose, and lignin). The thermal decomposition of biomasses in thermogravimetric experiments is described in the literature, and details of thermal decomposition processes are well known [31,32]. In the presence of air, sugar polymers, such as cellulose and xylan, start to be devolatilized between 200 °C and 350 °C, whereas lignin starts to decompose at 300–350 °C. These decompositions result in the formation of char, which is subsequently burnt at higher temperatures (between 400 °C and 500 °C). The temperature ranges of these processes strongly depend on both the nature of the biomass and the experimental conditions of TGA [33,34].

On the basis of the TGA results shown in Figure 1, three temperatures (250, 300, and 350 °C) were chosen for the thermal treatment of the investigated biomass samples in order to preserve the Ca oxalate phytoliths. The used experimental protocol was as follows: The different grapevine samples were weighed (about 9 g each), placed in a muffle furnace, and then heated at the desired temperature (250, 300, or 350 °C) for 3 h. At these low temperatures of calcination, the thermal residues still contained a high amount of carbon, looked like char, and could not be defined as ashes. In this temperature range, the devolatilization of sugar polymers occurs, which leads to the formation of char [31,32]. The resulting chars were then used for the characterization of the phytoliths.

### 2.3. Experimental Techniques Applied for Phytolith Characterization

2.3.1. Flame Atomic Absorption Spectroscopy (FAAS)

In order to obtain an accurate value for the content of Ca in the root, bark, and trunk samples, flame atomic absorption spectroscopy (FAAS) was used following hot-acid digestion in a mixture of $HNO_3$, $H_2O_2$, and $HCl$. The elemental quantification of Ca was performed using a PinAAcle 900F flame atomic spectrometer from PerkinElmer at a wavelength of 422.7 nm in an air–acetylene flame after acid digestion using microwaves.

2.3.2. X-ray Fluorescence (XRF)

Bulk elemental analysis was performed using wavelength-dispersive X-ray fluorescence (XRF) with a PANalytical Zetium (4kW) spectrometer. Prior to the analysis, 200 mg of washed and dried materials were milled with a Philips MiniMill planetary grinder. Both the jar and the four balls of the grinder consisted of zirconia. Subsequently, 200 mg of the resulting raw powder was pressed into 13 mm diameter pellets, with a pressure of 5 t. For the chars, 100 mg of each powder was mixed with 200 mg of boric acid and then pressed into 13 mm diameter pellets, with a pressure of 9 t.

2.3.3. X-ray Diffraction (XRD)

The XRD patterns of the raw, washed, and milled trunk, root, and bark and their chars were obtained on a PANalytical MPD X'Pert Pro diffractometer operating with Cu-K$\alpha$ radiation (K$\alpha$ = 0.15418 nm) and equipped with a PIXcel real-time multiple-strip detector (active length: 3.347°2θ). The powder XRD patterns were obtained at ambient temperature in the range 3 < °2θ < 70, a step of 0.013°2θ, and a data-collection time of 220 s/step.

2.3.4. Scanning Electron Microscopy (SEM)

Textural characterization was performed via scanning electron microscopy (SEM), using a JEOL JSM-7900F high-resolution microscope equipped with a BRUKER QUANTAX energy-dispersive X-ray (EDX) spectrometer. All images were taken in secondary electron (SE) mode. In order to investigate the calcium distribution in the various samples and more easily detect the Ca oxalate phytoliths, element distribution maps were generated, acquired at an accelerating voltage of 15 kV.

2.3.5. $^{13}$C-Nuclear Magnetic Resonance (NMR)

The $^{13}$C (I = 1/2) cross-polarization magic-angle spinning (CPMAS) NMR spectra were recorded at room temperature with a Bruker double-channel 4 mm probe on a Bruker Advance NEO 400 spectrometer, operating at B0 = 9.4 T (Larmor frequency $\nu_0$ ($^{13}$C) = 100.63 MHz and $\nu_0$ (1H) = 400.17 MHz). Samples were packed in a 4 mm diameter cylindrical zirconia rotor fitted with Kel-f end caps and spun at a spinning frequency of 12 kHz. $^{13}$C CPMAS NMR experiments were performed with a proton $\pi/2$ pulse duration of 3.3 μs, a contact time of 2 ms, and a recycle delay of 1 to 6 s depending on the 1H spin-lattice relaxation time (T1) measurements performed on each sample. In addition, 1H decoupled $^{13}$C MAS NMR experiments were recorded with a carbon $\pi/4$ pulse duration of 2.85 μs, a recycle delay of 60 s, and a 1 H high-power decoupling of 67 kHz. Chemical shifts reported below are relative to tetramethylsilane (TMS) using adamantane as the intermediate external reference.

## 3. Results and Discussion

The initial goal of this work was to try to find a method of extraction of Ca oxalate in order to quantify this organic salt in plants in all of their life stages. Two methods of extraction from Parr et al. (2001) were tested, and unfortunately, the liquid acid digestion led to the dissolution of this salt. Our hopes were then directed toward the dry extraction method of calcination. Calcination allowed for the thermal degradation of organic polymers from wood and led to an increase in the proportion of minerals in the residual chars.

Calcination was applied at low temperatures in order to ensure the recovery of Ca oxalate without thermally degrading it.

### 3.1. Bulk Chemical Composition of Plant Materials

In addition to the organic materials, the studied grapevine samples mainly contain Ca, K, Si, Al, Fe, Mg, and P (Table 1).

**Table 1.** Bulk chemical composition of the studied grapevine samples and their barks calcined at different temperatures. All values in wt%. Data for organic parts were obtained from TGA; all other data were obtained from XRF.

| | Trunk | Root | Bark | Bark Calcined at 250 °C | Bark Calcined at 300 °C | Bark Calcined at 350 °C |
|---|---|---|---|---|---|---|
| Organic component | 98.0 | 96.6 | 95.0 | 89.1 | 82.7 | 49.8 |
| $Na_2O$ | b.d.[a] | 0.05 | b.d.[a] | 0.16 | 0.57 | 1.62 |
| MgO | 0.08 | 0.08 | 0.17 | 1.08 | 1.77 | 4.61 |
| $Al_2O_3$ | 0.02 | 0.23 | 0.08 | 0.91 | 3.51 | 10.46 |
| $SiO_2$ | 0.04 | 0.41 | 0.17 | 0.92 | 3.71 | 11.68 |
| $P_2O_5$ | 0.04 | 0.2 | 0.02 | 0.08 | 0.55 | 1.55 |
| $K_2O$ | 0.44 | 0.20 | 0.31 | 1.00 | 1.15 | 2.89 |
| **CaO** | **0.34** | **1.33** | **3.81** | **6.89** | **6.29** | **18.48** |
| $TiO_2$ | 0.03 | 0.07 | b.d.[a] | b.d.[a] | 0.05 | 0.18 |
| $Cr_2O_3$ | b.d.[a] | b.d. | b.d.[a] | b.d.[a] | 0.08 | b.d.[a] |
| MnO | b.d.[a] | b.d. | b.d.[a] | 0.10 | 0.09 | 0.28 |
| $Fe_2O_3$ | b.d.[a] | 0.11 | 0.06 | 0.20 | 1.06 | 2.26 |
| Ni | b.d.[a] | b.d. | b.d.[a] | b.d.[a] | b.d.[a] | 0.02 |
| Cu | b.d.[a] | 0.008 | b.d.[a] | 0.03 | 0.04 | 0.08 |
| Zn | 0.003 | 0.009 | 0.006 | 0.02 | 0.08 | 0.23 |
| Rb | b.d.[a] | b.d.[a] | b.d.[a] | b.d.[a] | 0.005 | 0.009 |
| Sr | b.d.[a] | 0.004 | 0.009 | 0.02 | 0.02 | 0.05 |
| Pb | b.d.[a] | b.d. | b.d. | b.d.[a] | 0.006 | 0.008 |
| Zr | 0.78 | 0.36 | 0.30 | b.d.[a] | b.d.[a] | b.d.[a] |
| S | 0.02 | 0.08 | 0.04 | 0.14 | 0.27 | 0.71 |
| Cl | 0.01 | 0.009 | 0.02 | 0.04 | b.d.[a] | 0.08 |
| **Calcium oxalate monohydrate** [b] | **0.9** | **3.5** | **9.9** | **17.9** | **16.4** | **48.0** |

[a] b.d.: below detection. [b] amount of Ca oxalate calculated for the hypothetical case where 100% of Ca is present as this mineral species.

Calcium was found to be an abundant metal in all parts of the grapevine, whereby the CaO concentration, as determined using XRF, was highest in the bark (3.81wt%) and lowest in the trunk (0.34wt%). The concentrations of Ca determined using FAAS and expressed as CaO were similar, with values of 3.2%, 1.4%, and 0.28% for bark, root, and trunk, respectively. In addition to Ca, the main chemical element present, the studied grapevine samples also contained small amounts of various other elements (Table 1). The contents of inorganic components in different biomass types strongly depend on the nature and composition of the soils in which plants grow, and large compositional ranges have been observed [35]. The concentrations of the major and minor elements recorded for the three grapevine samples studied here were within the ranges reported in the literature for several wood types and energy crops [28,29,36].

The concentration of organic carbon in the grapevine, as determined using TGA, ranged from 95wt% in the bark to 98wt% in the trunk (Table 1). The difference to 100wt% represents the ash content, which thus ranged from 2wt% to 5wt% in the investigated samples, with the maximum observed for the bark (Table 1). The determined concentration of organic carbon in the grapevine samples is very similar to that reported for many

natural biomasses. Organic carbon is represented by polymers of sugars, lignin, and small molecules as extractables [37–41].

The observed maximum char content for the bark (Table 1) is consistent with the well-known fact that bark is the part of plants that exhibits the highest ash content [28]. Even if the bark represents only 3–5wt% of the total mass of wood, it contains most of the non-organic elements [28]. The main non-organic elements present in the bark are Ca, K, Mg, Na, Al, and Si, depending on both the nature of the wood and the chemical composition of the soil where the tree has grown. For example, Ca and K amounts in the bark for several types of wood range from 0.5wt% to 2.4 wt % and from 80 ppm to 0.7wt%, respectively [28,29].

Because of the high Ca content in the grapevine bark and because Ca oxalate crystals were found to be particularly abundant in this part of the plant (see below), Table 1 also lists XRF and TAG data for samples of the grapevine bark that were calcined at three different temperatures. The results document that, with increasing temperature, the organic content decreased substantially because the devolatilization of extractable compounds and the oxidation of sugars and lignin started to occur (see Figure 1). Consequently, calcination led to an increase in the ash content, with a concomitant increase in the concentration of inorganic components. The CaO content, for example, increased by a factor of nearly 3 between 250 °C and 350 °C.

### 3.2. Mineralogical Composition of Plant Materials

XRD was used to verify the purity of the purchased Ca oxalate reference material, labeled as $CaC_2O_4 \bullet xH_2O$, and to investigate the behavior of this compound during both the TGA and calcination processes at 250 °C. The diffractogram of the Ca oxalate reference matched well with the Ca oxalate monohydrate (Figure S2), in agreement with the TGA analysis (Figure 1). Traces of $Ca(OH)_2$ were also present in the purchased Ca oxalate in its initial form as well as after its calcination at 250 °C. As shown in Figure S2, the reference Ca oxalate was not affected by the thermal process at 250 °C.

XRD was further applied to identify the crystalline phases present in the different parts of the studied grapevine as well as in the bark samples after calcination. The XRD pattern of the trunk revealed that it did not contain detectable crystalline materials (Figure 2) except for small amounts of zirconia, which is indicated by relatively broad peaks (marked with * in Figure 2) and which represents contamination from the milling process.

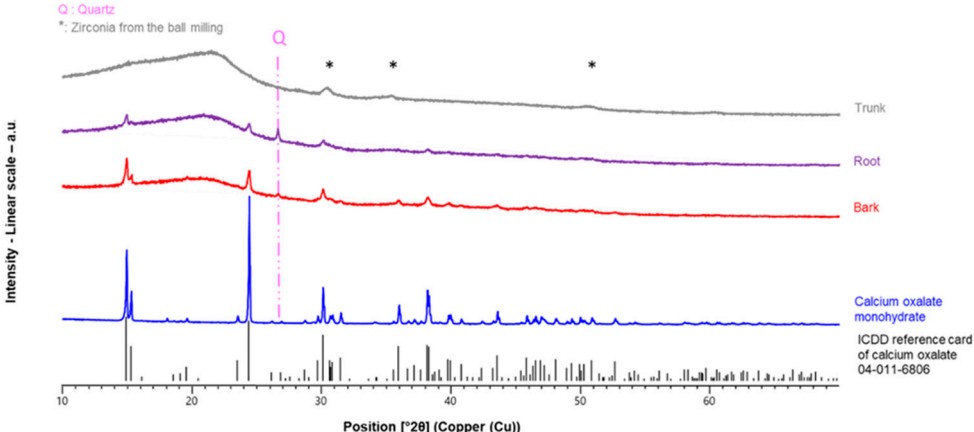

**Figure 2.** XRD patterns of the washed and milled samples of grapevine trunk, root, and bark, and of the purchased Ca oxalate monohydrate (blue line). The black vertical lines correspond to the XRD peaks of Ca oxalate monohydrate (International Centre for Diffraction Data (ICDD) Ref.# 04-011-6806).

On the other hand, crystalline Ca oxalate monohydrate, i.e., whewellite, was present in both the root and the bark of the studied grapevine, with larger amounts found in the bark. In addition, minor amounts of quartz were detected in the bark and the root (Figure 2). The characteristic quartz peak (highlighted in Figure 2) is more prominent in the XRD pattern

of the root than in that of the bark, consistent with the higher $SiO_2$ concentration recorded for the root (Table 1).

      The absence of calcite peaks in the XRD pattern of the studied original bark (Figures 2 and 3) might be due to a very low abundance of this mineral. As seen in Table 1, the original bark contained 3.81 wt% of CaO, which corresponds to 2.72 wt% of Ca. If all of this bulk Ca were present as either Ca oxalate monohydrate or calcium carbonate, it would correspond to 9.9 wt% or 6.8 wt% of these mineral species, respectively. It is well known that Ca salts in bark and wood can be a mixture of oxalate, carbonate, sulfate, phosphate, and organic acid groups, and that some of these minerals could also represent soil particles [28,35,42,43]. Even though the distribution of Ca between these different salts, and thus, the proportions of these salts, are not known in the studied bark sample, there is a good chance that calcite, $CaCO_3$, might be present in the bark, but in a concentration that would be below the detection threshold of the XRD analysis. If this were indeed the case, the calcite content could be elevated to detectable levels through calcination.

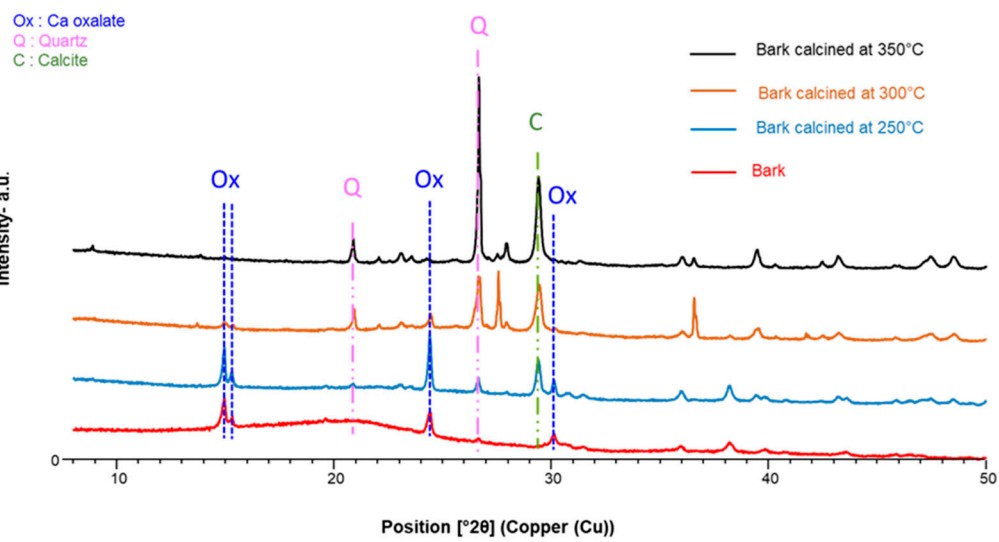

**Figure 3.** XRD patterns of the washed and milled grapevine bark and its calcination products generated at 250 °C, 300 °C, and 350 °C.

      Diffractograms of the calcined bark at the three calcination temperatures used in this study are presented in Figure 3. The calcination of the bark at 250 °C led to a slight increase in the XRD peaks of Ca oxalate monohydrate in the chars compared with the non-calcined natural bark (Figure 3), which corresponds to an increase in whewellite abundance. This enrichment was expected, because minerals that are originally present in a sample of biomass are concentrated in the associated char/ash, which has a total weight that is smaller than the initial weight of the precursor biomass sample. The XRD pattern further revealed the presence in the 250 °C ash of calcite, which was not detected in the original bark sample (Figure 3). Traces of quartz and kyanite were also detected.

      Calcination processes performed at a higher temperature revealed that Ca oxalate monohydrate underwent decomposition, as documented by the heights of its XRD peaks, which are smaller at 300 °C than those seen at 250 °C, and which are no longer visible at 350 °C (Figure 3). At the same time, the XRD peaks of calcite become more prominent with increasing temperature, and at 350 °C, calcite is—along with quartz—the main crystalline component in the bark. Other crystalline phases detected at 350 °C include traces of feldspar and muscovite.

      The TGA data (Figure 1) revealed that the reference Ca oxalate monohydrate started to decompose at 400 °C with the loss of one molecule of CO to form calcite (Reaction 2). These results are in agreement with previous TGA experiments performed on Ca oxalate [30]. In contrast, calcite already appeared at 250 °C during the calcination of the

grapevine bark in a muffle furnace (Figure 3). This result might indicate that, in a muffle furnace, where the sample was exposed to static air conditions, Reaction 2 started at a lower temperature than that during the TGA performed under an airflow. To test this hypothesis, the reference material of Ca oxalate monohydrate was calcined at 250 °C in the same muffle furnace under the same conditions as those used for the calcination of the biomass samples. The superposition of the diffractograms of Ca oxalate monohydrate and its residue after calcination in the muffle furnace (Figure S2) shows that the two materials are almost identical and that calcite is not detectable in the calcination product. The only differences between the two XRD patterns are a slight broadening of the peaks, which suggests minor amorphization, and an increase in background noise. These experimental data document that Ca oxalate monohydrate did not decompose to calcite at 250 °C under the static air conditions of the muffle furnace.

A recent publication may explain why $CaCO_3$ is formed during calcination at a temperature below 450 °C: Using TGA coupled with mass spectrometry, it was demonstrated that three additional reactions exist during the thermal degradation of Ca oxalate monohydrate [44]. The decomposition of Ca oxalate monohydrate in an inert atmosphere and in the presence of low amounts of oxygen is more complex than anticipated and consists of the following cascading reactions [44], which are superimposed onto the well-known [30,45] thermal decomposition Retractions (2) and (3):

$$Ca(C_2O_4) \rightarrow CaCO_3 + 0.5\ CO_2 + 0.5\ C \tag{4}$$

$$0.5\ C + 0.5\ CO_2 \rightarrow CO \tag{5}$$

$$0.5\ CaCO_3 + 0.5\ C \rightarrow 0.5\ CaO + CO \tag{6}$$

The inspection of C, CO, and $CO_2$ profiles at masses $m/z$ = 12, 28, and 44, respectively, in the paper by Hourlier (2019) revealed that the emission of $CO_2$ starts during dehydration, but that most of C, CO, and $CO_2$ compounds are produced during the thermal phases of the formation of $CaCO_3$ through Reaction (2) and decarbonation through Reaction (3). Reaction (4) was observed during the temperature range corresponding to Reaction (2) and, as documented by its negative Gibbs free energy, was thermodynamically favorable regardless of temperature. The relationships between the Gibbs free energies of all these reactions showed that Reactions (2) and (4) could occur [44]. Reaction (5), known as the Boudouard reaction, also occurs during the temperature range of Reaction (2). The simultaneous occurrence of C, $CO_2$, and CO was also observed at higher temperatures, during the decarbonation of $CaCO_3$, which can be explained by the superposition of Reaction (6) onto Reaction (3).

### 3.3. Characterization of Individual Phytoliths

The SEM images of the washed grapevine samples (Figures 4, 5 and S3) document the close association of minerals with the organic matrix mentioned above.

Most of the crystals detected in both the root and the bark were Ca oxalate crystals, as suggested by EDX spectra and X-ray distribution maps (see, for example, Figures 4b,d and 5b). The XRD patterns of the bark and the root revealed that these Ca oxalate crystals are the monohydrate form whewellite (Figure 2). Other phases were observed as well (e.g., Figure 4b,d), but they were present only in small quantities or trace amounts. These minerals include rounded grains of quartz, consistent with the XRD data (Figure 2), as well as flakes of aluminum silicates. Based on their habit, these Al silicates are most likely clay minerals, but unequivocal identification was not possible, because they occurred in such low abundance that they did not generate characteristic XRD peaks (Figure 2).

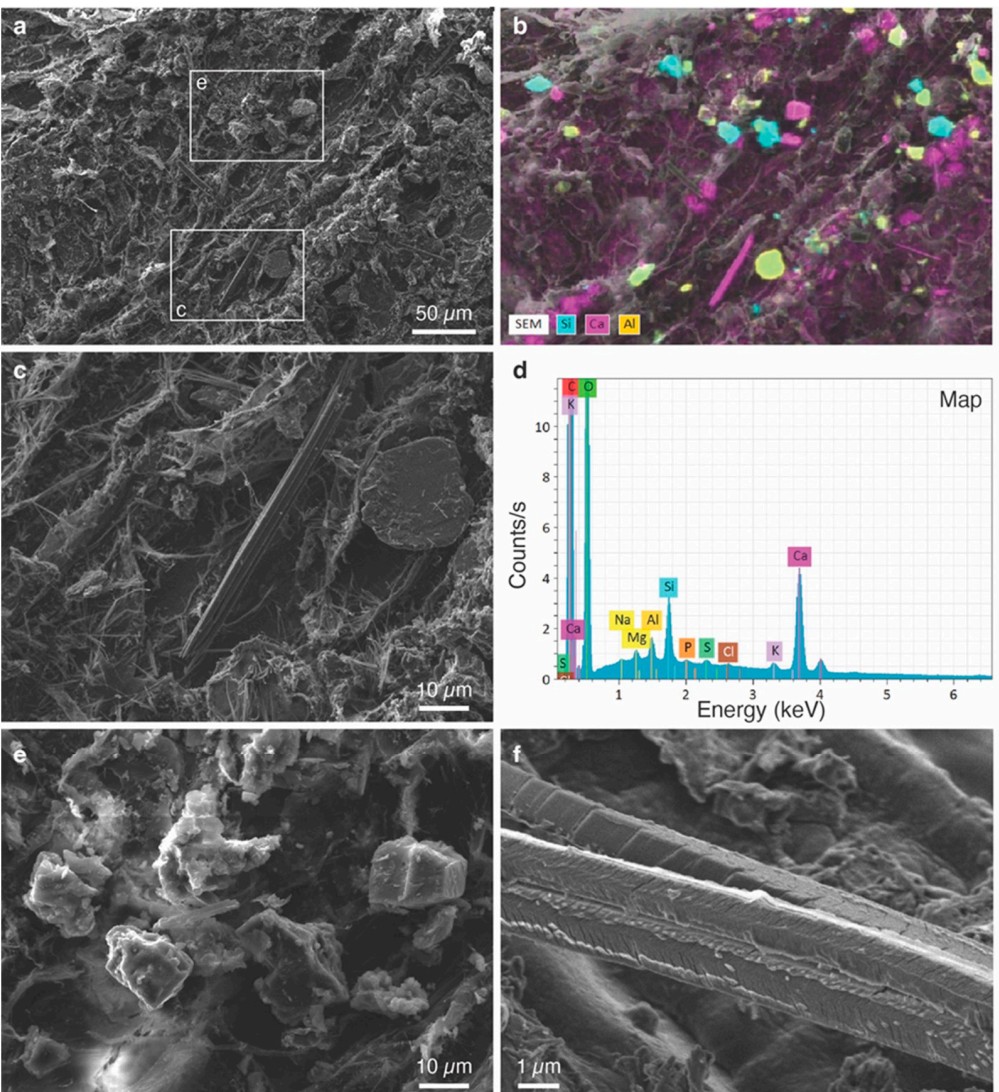

**Figure 4.** Washed grapevine root: (**a**) SEM image showing a typical overview with various mineral grains embedded in the organic plant structure; white rectangles outline the areas shown in (**c**,**e**); (**b**) X-ray distribution maps of Si, Ca, and Al superimposed onto the SEM image shown in (**a**) and visualizing different types of minerals: blue = quartz; magenta = Ca oxalate, and green = minerals containing both Si and Al; (**c**) detail of (**a**) showing a needle-shaped Ca oxalate crystal, consisting of several needles, and an isometric Si–Al-containing mineral, whose shape suggests that it represents a clay mineral from the soil; (**d**) EDX spectrum revealing the presence of the most abundant elements in the area mapped in (**b**); (**e**) detail of (**a**) showing two isometric Ca oxalate crystals and two quartz crystals; (**f**) SEM image of two acicular, twinned Ca oxalate crystals.

The whewellite crystals were typically euhedral. In the grapevine root, many crystals were needle-shaped or bladed, exhibiting sharply pointed ends and grooves along their sides (Figure 4c), and thus, they were typical raphides. These acicular crystals were up to 75 μm in length. Some of the raphide crystals in the root were twinned along their length and segmented (Figure 4f), showing features similar to those described previously for grapevine leaves [19]. Other whewellite crystals in the root were equant or short-prismatic and, in some cases, bipyramidal. Grain sizes of individual non-raphide crystals, which could also be twinned, ranged between 3 and 18 μm across, with an average of $10 \pm 4$ μm ($n$ = 21). Smaller crystals preferentially occurred as crystal sand.

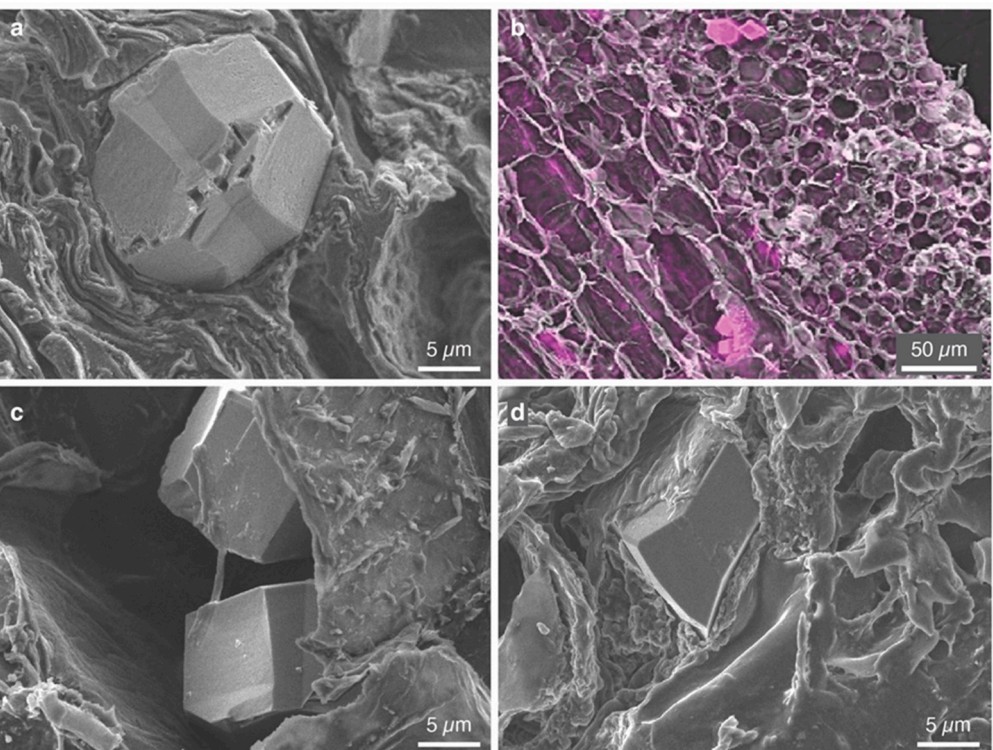

**Figure 5.** Washed grapevine bark: (**a**,**c**,**d**) SEM images of Ca oxalate crystals; (**b**) X-ray distribution map of Ca superimposed onto a corresponding SEM image and documenting the presence of Ca oxalate crystals, including those displayed in (**a**,**c**).

In the bark of the studied grapevine, the whewellite crystals exhibited equant or short-prismatic habits (Figure 5); no raphides were observed. The crystals ranged in size between 9 and 27 μm across, with an average grain size of 16 ± 4 μm (*n* = 41). In many cases, the whewellite crystals were twinned (e.g., Figure 5a).

In the trunk of the studied grapevine, whewellite was present as short-prismatic, bipyramidal, or rhombohedral crystals, but also as long prisms (Figure S3). The prismatic crystals were frequently twinned. No raphides were found. Crystal size varied from 3 μm to 28 μm, and the average grain size was 11 ± 7 μm (*n* = 19), whereby the smallest crystals were observed as crystal sand. The calcium oxalate crystals showed the typical habits and twins reported in the literature [46].

As a result of calcination, the Ca oxalate crystals in the bark were partially altered. Alteration features included surface modifications and cracking (Figure 6), but the grain size did not seem to be altered. Unlike the surfaces of the Ca oxalate crystals in the original bark (Figure 5a,c,d), the whewellite surfaces after calcination were characterized by an abundance of rounded to elongate pores (50–200 nm across). The modified surfaces can thus be classified as macroporous. Both the porous crystal surfaces and the cracks could be seen already at 250 °C but were especially well visible after calcination at 300 °C and 350 °C, when most of the surrounding organic material had been removed (Figure 6). Larger, elongate openings with tabular shapes and straight borders were also observed (longest dimension varying between 0.5 and 1.7 μm); they were generally arranged parallel to prominent crystal faces and were particularly prominent at 350 °C (Figure 6f).

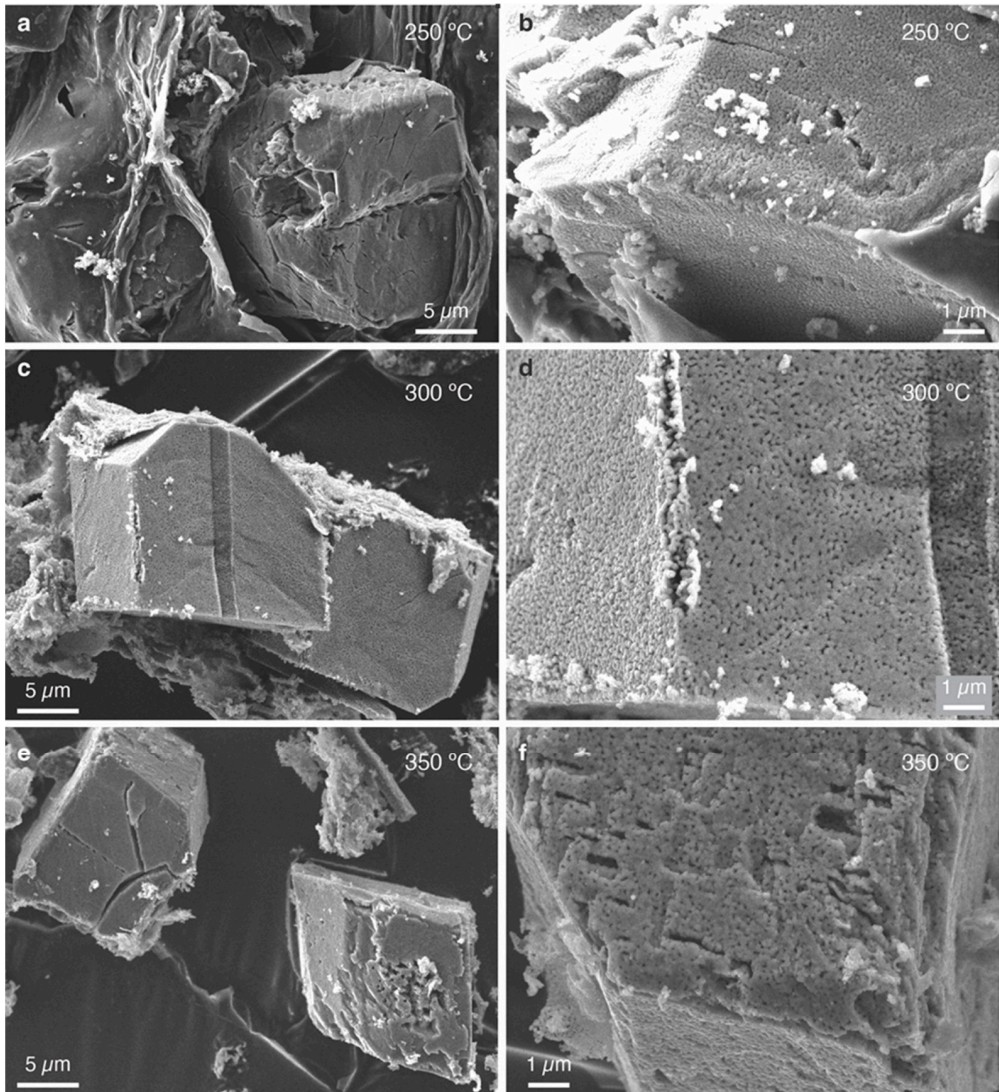

**Figure 6.** SEM images of grapevine bark calcined at different temperatures: (**a,b**) bark calcined at 250 °C; (**c,d**) bark calcined at 300 °C; (**e,f**) bark calcined at 350 °C.

The Ca oxalate crystals in the root underwent analogous surface modifications and cracking as a result of calcination at 250 °C (Figure S4). The calcination of the grapevine root at higher temperatures was not attempted.

The presence of macropores, tabular openings, and cracks on the surface of the Ca oxalate crystals after calcination suggests that these features may have been formed as a result of the release of $H_2O$ and CO during the thermal degradation of the original whewellite crystals, and thus, that they represent escape paths for volatiles.

### 3.4. $^{13}$C-NMR Analysis

The carbon-containing components of the plant materials were identified using $^{13}$C CPMAS NMR (Figure 7).

According to Wooten (1995), Ca oxalate monohydrate exhibits a massif centered at 168 ppm. This shift was also seen in the NMR spectrum of the purchased Ca oxalate [47]. The NMR spectra of the studied biomass samples reveal traces of Ca oxalate, mainly in the bark (red line in Figure 7).

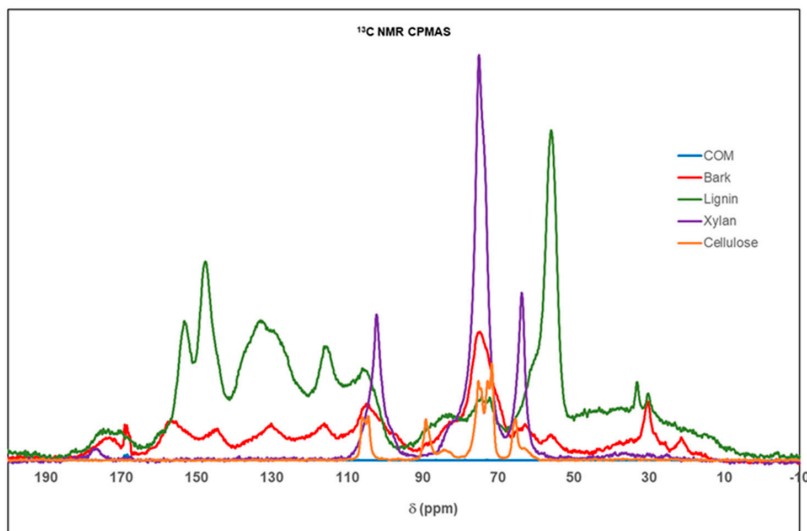

**Figure 7.** [13]C CPMAS NMR spectra of the purchased Ca oxalate monohydrate (COM) bark from the grapevine studied here, and of purchased lignocellulosic polymers.

The [13]C-NMR spectrum of the studied bark also shows several broad resonances at 22 ppm, 35 ppm, 75 ppm, 105 ppm, 115 ppm, 130 ppm, 145 ppm, and 157 ppm, which correspond to the organic compounds present in the biomass, as shown in the spectra of the reference materials lignin, cellulose, and xylan (Figure 7). The broad signal centered at 75 ppm in the bark spectrum is attributed to carbohydrates from sugar polymers cellulose and xylan [47], whereas all other resonances confirm the presence of lignin. Signals below 50 ppm correspond to aliphatic functions from lignin. In the range of 100 to 160 ppm, large peaks characterize aromatic functions from lignin [47].

The [13]C-NMR MAS + DEC spectrum of the calcined bark (Figure 8) reveals the presence of Ca oxalate, whereby an increase in the intensity of the signal relative to that of the original biomass material (red line in Figure 8) was observed, especially for the bark calcined at 250 °C. Ca oxalate monohydrate showed chemical shifts at 167.6, 169.3, and 169.0, respectively. The spectrum still contained resonances of biomass polymers.

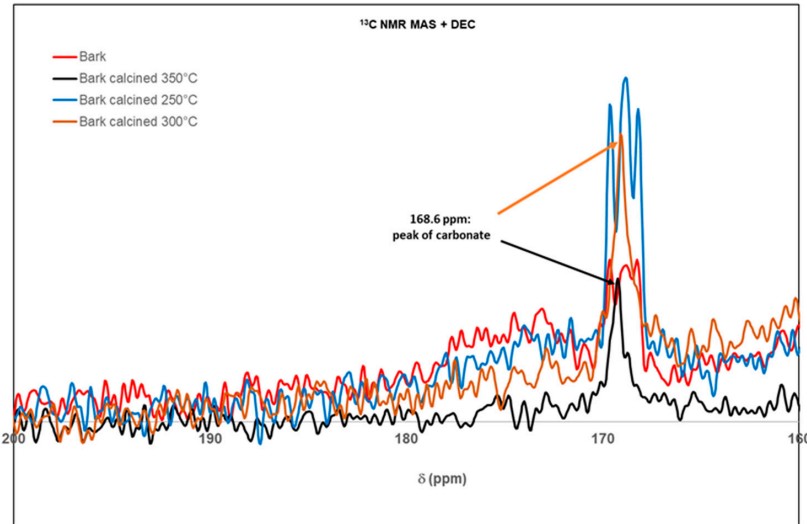

**Figure 8.** Detail of the [13]C-NMR MAS + DEC spectra showing the oxalate region (between 160 and 180 ppm) for the original grapevine bark and the residual chars after calcination at various temperatures (entire spectra shown in Figure S5).

As shown in Figure 1, biomass polymers decomposed in the temperature range of 250 °C to 430 °C. Consequently, at 250 °C, the thermal decomposition just started and was not yet completed. In Figure 8, which represents a zoom of the region from 160 to 200 ppm, a peak at 168.6 ppm is observed at 300 °C but not at lower temperatures. This peak, which overlaps with those of Ca oxalate, corresponds to Ca carbonate, but it was not observed with the $^{13}$C CPMAS NMR technique, probably because of the absence of protons in close vicinity.

In agreement with the XRD data, the $^{13}$C-NMR spectra confirm that Ca oxalate started to decompose at a temperature of 300 °C, producing Ca carbonate by releasing one molecule of carbon monoxide. This fact could be explained by Reaction (4), which starts during the dehydration of the whewellite crystals.

## 4. Conclusions

This study revealed that, in order to characterize Ca phytoliths from grapevines, calcination could be performed at temperatures below 300 °C. This procedure favors the enrichment of minerals in residual chars, which then can be characterized using SEM, XRD, and $^{13}$C-NMR. The procedure, however, did not allow for the complete extraction of phytoliths via calcination from the grapevine due to the formation of chars in the studied temperature range. At temperatures around 300 °C, Ca oxalate decomposed into calcite. The applied dry extraction technique did not allow for the quantification of Ca phytoliths in the hardwood parts of the studied grapevine (trunk, bark, and root), because the chars contained several other minerals that could not be separated. The next step of this research project, now in progress, involves an examination of the leaves of the same grapevine species after the harvest period in autumn by applying the calcination protocol. This paper demonstrates the value of using a combination of bulk analytical techniques, including NMR spectroscopy and XRD, to identify the mineral components present in biomass.

**Supplementary Materials:** The following supporting information can be downloaded at: https://www.mdpi.com/article/10.3390/cryst13060967/s1, Figure S1: Photograph showing the various parts of the studied grapevine (*Vitis vinifera*) and their length. Yellow dashed lines indicate where the parts were separated from each other; Figure S2: XRD patterns of the purchased Ca oxalate material and its residual after calcination at 250 °C in a muffle furnace. The blue vertical lines correspond to the XRD peaks of Ca oxalate monohydrate (International Centre for Diffraction Data (ICDD) Ref.# 04-011-6806), the tan-colored vertical lines to the peaks of Ca hydroxide (ICDD Ref.# 01-076-0571), Figure S3: Washed grapevine trunk. SEM images showing (a) a prismatic Ca oxalate crystal; (b) two rhombohedral Ca oxalate crystals embedded in the organic matrix of the trunk, Figure S4: SEM image of grapevine root calcined at 250 °C showing Ca oxalate crystals with the typical surface modifications, which are analogous to those observed for calcined grapevine bark (Figure 6 in the main text), Figure S5: $^{13}$C NMR MAS + DEC spectra of the grapevine bark and of the residual ashes after calcination at various temperatures.

**Author Contributions:** All authors contributed to the conceptualization and design of this study. Material preparation, data collection, and analysis were performed by D.K., L.J., L.M. and S.R. Conceptualization and methodology were performed by B.L. and G.T. The first draft of the manuscript was written by G.T. and R.G. and all authors commented and improved successive versions of the manuscript. All authors have read and agreed to the published version of the manuscript.

**Funding:** This work was supported by funds from the Université de Haute-Alsace, (grant numbers UHA/TROUVE/PIR2022).

**Data Availability Statement:** Not applicable.

**Acknowledgments:** Authors thank C. Bertsch and S. Farine from the Laboratory LVBE (Laboratoire Vigne, Biotechnologies et Environnement) of UHA for their technical support in regard to grapevine sampling. We also thank Ines Hasnaoui for experimental assistance from the University of Strasbourg. The authors thank the four reviewers for their suggestions to improve the manuscript.

**Conflicts of Interest:** The authors declare no conflict of interest.

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
