# Peer review of "The Multi-Analytical Characterization of Calcium Oxalate Phytolith Crystals from Grapevine after Treatment with Calcination"

_crystals, doi:10.3390/cryst13060967_

Round 1

Reviewer 1 Report

The Manuscript entitled "Multi-characterization of calcium-oxalate phytoliths crystals from grapevine after treatment by calcination" describes different techniques and characterization of crystalline calcium-oxalate monohydrate.

I believe the work is well presented and documented. However, this reviewer thinks the introduction can be improved by providing reported methods to investigate calcium phytoliths. Also, the importance of the current study and its future applications.

Author Response

  • The Manuscript entitled "Multi-characterization of calcium-oxalate phytoliths crystals from grapevine after treatment by calcination" describes different techniques and characterization of crystalline calcium-oxalate monohydrate.

I believe the work is well presented and documented. However, this reviewer thinks the introduction can be improved by providing reported methods to investigate calcium phytoliths. Also, the importance of the current study and its future applications.

 We have now expanded the introduction by including a sentence on the importance of phytoliths in various types of research. We also moved a paragraph from the experimental part to the introduction to discuss several extraction protocols of phytoliths and added a new reference on the subject (Parr and Sullivan, 2014).

  • Does the introduction provide sufficient background and include all relevant references?

Yes, now we have included additional references in the introduction and explained why phytoliths are important. Four new references were added.

  • Are all the cited references relevant to the research?

In our opinion, yes! We even added some more, in response to reviewer requests.

Reviewer 2 Report

Submitted manuscript demonstrates the value of using a combination of bulk analytical techniques, that is NMR and XRD spectroscopy to identify minerals present in biomass. The work has been prepared in a transparent and clear way.

I believe that Figures 7-8 presenting the 13CNMR spectras should be enlarged to make it more readable. Please check the correctness of the English language.

Author Response

  • I believe that Figures 7-8 presenting the 13CNMR spectras should be enlarged to make it more readable. Please check the correctness of the English language.

All figures were enlarged. The manuscript was read and corrected by a native English speaker.

Reviewer 3 Report

The paper is within the aims and the scope of the journal. Methods are suitable, and used in a way that is possible to replicate experiments and analyses. The investigation is performed to good technical standards. It is no ethical problem involved. Discussion is sound and relevant. Conclusions should be extended.

 Suggestions:

Table 1. Improve the first (left) column, first and last row in the Table.

Give more details on the method of washing plant parts.

Author Response

  • Does the introduction provide sufficient background and include all relevant references?

Yes, now we have included additional references in the introduction and explained why phytoliths are important. We also moved a paragraph from the experimental part to the introduction to discuss several extraction protocols of phytoliths and added a new reference on the subject (Parr and Sullivan, 2014). Four new references were added.

  • Table 1. Improve the first (left) column, first and last row in the Table.

Thank you. The problem has been corrected!

  • Give more details on the method of washing plant parts.

A sentence was added in the part 2.1 Starting materials as follows: “The grapevine sample was washed with large volumes of water for 15 min to remove soil particles and subsequently sun dried outside for two weeks”.

  • Does the introduction provide sufficient background and include all relevant references?

Yes, now we have included additional references in the introduction and explained why phytoliths are important. We also moved a paragraph from the experimental part to the introduction to discuss several extraction protocols of phytoliths and added a new reference on the subject (Parr and Sullivan, 2014). Four new references were added.

  • Are the methods adequately described?

Concerning the experimental procedure of extraction of Ca-Phytoliths, the authors believe that they are sufficiently described because they are the main objective (and subject) of the paper. Concerning characterization of Ca-Phytoliths (13C-NMR, XRD, SEM, etc.), the methods used conform with what has been done in other publications.

Reviewer 4 Report

This is quite interesting paper, dealed with characterization of phytrolites of vineyard plant. It is good work in terms of mineralogy and chemistry, several instrumental methods are used for complex characterisation of the phytolites. Few minor suggestions are given below. One general comment - it is nessesary to discuss environmental aspects of phytolites investigation, thus your methodic has paleogeographical interpretation, e.g. work of A.A. Golyeva from Isntitute of Geography, Moscow, who have used the same type of phyrolites for geochronological research. Thus, I reccomen at list cite this type of works and to discuss environmental aspects of phytolites investigation.

Vitis vinifera Chasselas - provide it in italic, as normal latin name

This tpye of references are not clear:  Phyllis2. (2022a). Wood, Beech (#797). 621 39. Phyllis2. (2022b). Wood, Beech (#2142). 622 40. Phyllis2. (2022c). Wood, Fir (#791. 623 41. Phyllis2. (2022d). Wood, Spruce (#163). 624 42. Phyllis2. (2022e). Wood, Spruce (#2402

Author Response

  • This is quite interesting paper, dealed with characterization of phytrolites of vineyard plant. It is good work in terms of mineralogy and chemistry, several instrumental methods are used for complex characterisation of the phytolites. Few minor suggestions are given below. One general comment - it is nessesary to discuss environmental aspects of phytolites investigation, thus your methodic has paleogeographical interpretation, e.g. work of A.A. Golyeva from Isntitute of Geography, Moscow, who have used the same type of phyrolites for geochronological research. Thus, I reccomen at list cite this type of works and to discuss environmental aspects of phytolites investigation.

Golyeva studied mainly opaline phytoliths, which are not the topic of this paper. However, we added a sentence (with some key references) to explain in which types of research the study of phytoliths is important, for example in soil science and paleoecology (see Introduction). To our knowledge, this is the research direction that A.A. Golyeva had started in 1997 and which has expanded over the past two decades.

  • Vitis vinifera Chasselas - provide it in italic, as normal latin name

Corrected everywhere in the text and in the figure captions, thank you!

  • This tpye of references are not clear:  Phyllis2. (2022a). Wood, Beech (#797). 621 39. Phyllis2. (2022b). Wood, Beech (#2142). 622 40. Phyllis2. (2022c). Wood, Fir (#791. 623 41. Phyllis2. (2022d). Wood, Spruce (#163). 624 42. Phyllis2. (2022e). Wood, Spruce (#2402

The Phyllis2 is an European database for biomasses in which chemical and thermal properties can be found. The references were modified and amended with additional details.

  • Are all the cited references relevant to the research?

Yes, now we have included additional references in the introduction and explained why phytoliths are important. We also moved a paragraph of the experimental part from the introduction to discuss several extraction protocols of phytoliths and added a new reference on the subject (Parr and Sullivan, 2014). Four new references were added.
